# Tapping the Employee Perspective on the Improvement of Sustainable Employability (SE): Validation of the MAastricht Instrument for SE (MAISE-NL)

**DOI:** 10.3390/ijerph17072211

**Published:** 2020-03-25

**Authors:** Inge Houkes, Massimo Miglioretti, Eleonora Picco, Angelique Eveline De Rijk

**Affiliations:** 1Department of Social Medicine, CAPHRI Care and Public Health Research Institute, Faculty of Health, Medicine and Life Sciences, Maastricht University, 6200 MD Maastricht, The Netherlands; 2Department of Psychology, University of Milano-Bicocca, 20126 Milano, Italy; 3BiCApP, Bicocca Center for Applied Psychology, University of Milano-Bicocca, 20126 Milano, Italy

**Keywords:** vitality, employees, well-being, questionnaire, psychometric properties, sustainable employability

## Abstract

*Background:* Sustainable employability (SE) is top priority for employers. Measures based on the employee perspective of SE that would give direction to interventions to be initiated by employers currently fall short. This study aims to develop and validate an instrument that captures these issues: the MAastricht Instrument for Sustainable Employability (MAISE). *Methods:* MAISE items were generated from an extensive literature review and interviews with employers and employees. A questionnaire containing these items as well as proxy variables (health and vitality) and demographics was answered online by 632 employees (response rate 50.3%). Construct validity, reliability, and criterion validity were tested through Principal Component Analysis (PCA), Confirmatory Factor Analysis (CFA), Cronbach’s alpha, and correlational analyses. *Results:* The MAISE comprises 12 scales divided over five areas: (1) Meaning of SE; (2) Level of SE; (3) Factors affecting my SE; (4) Overall responsibility for SE; and (5) Responsibility for factors affecting my SE. Reliability, construct, and criterion validity were adequate to good. SE of the employees was relatively high, and SE was considered a shared responsibility of the employee and employer. *Conclusions:* This study showed the MAISE to be reliable and valid in various employee groups. More validation studies are needed. We recommend that employers use the MAISE as a needs assessment in order to develop SE interventions that will be readily accepted.

## 1. Introduction

As a consequence of an ageing labor population, keeping employees at work for as long as possible and as fit as possible is top priority for employers. It is in this context that the construct of sustainable employability (SE) has been introduced. However, there are several lacunae in the debate and research on SE [1,2]. In this paper, we focus on the gap between conceptual and intervention issues. We argue that the employee perspective on SE might fill this gap and has not yet received sufficient attention. Next, we propose a new instrument which builds upon recent scientific developments, taps the employee perspective on SE, and gives direction to interventions.

### 1.1. The Construct of SE

Employability was initially defined as a person’s ability to gain and maintain employment and to be productive [3]. More recently, it has been defined as “a personal resource that individuals develop across their working lives aimed at increasing one’s own career success, both attaching importance to (i.e., employability orientation) and committing to (i.e., employability activities) making sense of past work experiences and envisioning one’s own professional future, acquiring valuable competencies and skills, improving their formal and informal career-related networks, exploring their social environment in search of opportunities and constraints to their own career pathway” [4] (p. 196). The adjective “sustainable” introduces the time horizon of pension age as well as the well-being aspect of employment. Recently, the notion that working life contributes to individual values has been added to the construct of sustainable employability. Inspired by Amartya Sen’s Capability approach [5], it is assumed that both work factors and individual factors can lead to a set of capabilities (potentials) to function well at work. This is only achieved if personal and contextual factors that can convert work and individual characteristics into capabilities are present [2]. Hence, sustainable employability is not a personal characteristic but rather the result of an interaction among the individual, his or her job, and the social/organizational context in which it is put into practice [6,7,8]. The employee brings in health, personality, as well as emotions and motivations, where the job is a combination of specific tasks along with cultural aspects and development possibilities within a specific organization. The current conceptualization of sustainable employability comprises four components: the health component (e.g., well-being, vitality, and quality of working life); the productivity component (e.g., work ability, productivity, work engagement, and work performance); the valuable work component (e.g., positive attitude, job motivation, and having the right competences for one’s work); and, considering the long-term goal of SE, the component of a long-term perspective [6]. Thus, an employee who is physically and mentally healthy, has the competencies that fit the job, is motivated and engaged, and expects to be able to work productively until retirement, can be considered sustainably employable. Thus, the SE construct is related to, but not similar to employee engagement and work ethic [9,10].

### 1.2. Interventions for SE

Interventions to promote SE also vary, but not along the same lines as the definitions. The distinction is rather between individual interventions such as lifestyle training [11] and training of employee skills aimed at solving problems such as ageing and chronic health conditions [11,12] and interventions that focus on changes in the organization of the work. Van Vuuren and Van Dam [13] have distinguished the latter, less frequently seen interventions into four subtypes: (1) work adaptations (extension or change of duties and more autonomy); (2) increase in employability by job rotation and internships, job-related education, and training; (3) the “New Way of Working” with flexible working times and working from home; and (4) a positive working environment characterized by coaching leadership styles and a positive working culture. However, the evidence base for SE interventions is small. Hazelzet et al. [6] reviewed SE interventions and included only those studies in which interventions were presented as SE interventions by the authors themselves. They showed that the number of studies is low, that the quality of the evidence base is weak to moderate, and that the effectiveness of interventions is at best ambiguous. Considering the content of the interventions evaluated, at least two SE core components, “health” and “valuable work”, were addressed in all interventions evaluated; the productivity and long-term components were addressed to a much lesser extent. The authors partly explain the limited effectiveness by the inconsistencies in the operationalization of outcome measures and the lack of alignment between SE conceptualization, intervention content, and effect measurement. Decisions on how to intervene seem not to be based on the literature or evidence but rather on the perspectives of the researcher [11] and employer [13] instead of the perspective of the employee.

### 1.3. Need for an Employee Perspective on SE and Pointers for Intervention Development

In most conceptualizations of SE and SE interventions, the employer’s perspective has been the focus. In line with the Capability approach but from a different starting point, we argue that the employee perspective has been missing from conceptualization and measurement of SE and, thus, in the choice of interventions. Horstman [14] called for more space for employees to give meaning to their own SE and for giving employees more voice in promoting their SE. In a similar vein and in line with democratic thinking, most researchers and employers regard both the employer and the employee as being responsible for SE [15]. If we assume that SE is a multidimensional construct (consisting of several core components), opinions on responsibility may vary along with the dimensions. Additionally, employees in different contexts may well have different perspectives on SE. These notions have implications for how SE should be measured.

### 1.4. Measuring SE 

Until relatively recently, SE has mainly been measured by means of a range of proxies such as health, vitality, work ability, motivation, productivity, sickness absence, and the accompanying scales and instruments [7]. This research area focuses on the well-known concepts related to healthy work such as autonomy, rewards, engagement, and competence and puts these concepts into a longitudinal and/or age-related perspective [16,17,18]. Similarly, Fleuren et al. [1] proposed studying the full causal model, the inclusion of both determinants of SE relating to the job and the employee, and SE itself as an outcome over time. A second area of research focuses on the attitudes towards working until pension age [19]. More recently, two specific SE measures have been developed. First, there is the Vitality scan [8]. This instrument comprises five scales (i.e., balance and competence, motivation and involvement, resilience, mental and physical health, and social support at work). The Vitality scan has a moderate to good reliability and construct validity. The criterion validity of this instrument has not yet been examined [8]. Second, the Capability set for work has been introduced [2,20]. This is a core set of seven capabilities comprising use of knowledge and skills, development of knowledge and skills, involvement in important decisions, building and maintaining meaningful contacts at work, setting one’s own goals, having a good income, and contributing to something valuable. An employees’ SE is determined by an algorithm. This questionnaire appeared to be a valid instrument to measure a worker’s capability set, but criterion validity was relatively low: significant but low correlations were found between the capability set and work outcomes such as work functioning, work ability, and sickness absence [20]. Both the Vitality scan and the Capability set have merits and advantages, particularly for monitoring the individual course of SE and for intervening at an individual level. In order to create a stronger evidence base for the effectiveness of SE interventions, we argue that an instrument is needed which measures the core components of SE and taps the different meanings and factors associated with SE from an employee perspective and the responsibility for SE. Such information will provide a needs assessment for employers which will facilitate their decision-making process and the development of interventions for (subgroups of) employees. 

### 1.5. Aim of this Study and Research Questions

The aim of this study is to develop an instrument for SE that incorporates the diverse factors that affect SE and those experiences of SE that reflect the employee perspective and notions of responsibility. Research questions were the following: How (i.e., by means of which scales) can SE, factors affecting SE, division of responsibility for SE, and responsibility for factors affecting SE be measured from an employees’ perspective?What are the reliability and validity of these scales?

## 2. Materials and Methods 

### 2.1. Development of the MAastricht Instrument for Sustainable Employability

The MAastricht Instrument for Sustainable Employability (MAISE) was developed in three steps. First, a literature review was performed. We included research papers and reviews [7,21,22] and decided to include practice reports about sustainable employment [23,24]. For an extensive description of this literature review and the development of MAISE, see Houkes et al. [11]. Based on this review, a preliminary list of aspects relevant to SE was developed. Second, unstructured individual interviews addressing perspectives on SE (Autumn 2014) were performed with various stakeholders (two Human Resource managers, one organizational adviser, two occupational physicians, and two employee representatives). These interviews were transcribed and analysed, and the results of these analyses were used as input for one focus group meeting (representing the same stakeholder groups but consisting of different people (*n* = 7)), during which we tried to reach consensus about the most important aspects of SE and to get more insight into both the employer and employee perspectives on SE. This led to another set of aspects relevant to SE, which was compared and combined with the preliminary set which resulted from the literature review. This resulted in a comprehensive list of themes relevant to the meaning and promotion of sustainable employability, including health, vitality, lifestyle issues, career and growth opportunities, using skills and knowledge, social climate, role of supervisor, feeling appreciated, feeling challenged, reasonable workload and work pressure, willingness to work longer, willingness to be educated, willingness to change job and to take on new tasks, organizational adjustments, pleasure in work, satisfaction, organizational culture, flexible working arrangements, work–life balance, and HR management. Third, these themes were clustered and a list of items was developed and discussed by the authors (I.H. and A.E.d.R.) and two additional researchers (M.B. and A.K.; see Acknowledgments), all of whom are experts in the fields of work and health and SE. Several existing questionnaires such as the Work Limitations Questionnaire [25] and the Vitality scan [8] were screened in this process, but scales and items were not directly copied from existing questionnaires.

This process resulted in a final set of items organized into five areas, all of which are measured from an employee perspective:The meaning of SE according to the employee (10 items). The set starts with “Sustainable employability has the following meaning to me:”. An example item is “The capacity to do my job efficiently”. The response scale ranges from 1 “Strongly disagree” to 5 “Strongly agree”.The level of SE of the employees themselves (10 items). This set starts with “To what extent do the following statements apply to you?”. An example item is “I am efficient at my job”. The response scale ranges from 1 “Strongly disagree” to 5 “Strongly agree”.Factors affecting SE of the employee (18 items). This set starts with “Indicate to which extent you believe the following changes could affect your sustainable employability:”. An example item is: “Improvement of working conditions”. The response scale ranges from 1 “Not at all” to 5 “A huge amount”.Responsibility for SE (1 item, “With whom does the responsibility for sustainable employability lie according to you?”. The response scale ranges from 1 “Only with the employer” to 5 “Only with the employee”.Responsibility for factors affecting SE (18 items). This set starts with “Indicate where you feel the responsibility lies for implementing the changes below that would improve your sustainable employability”. These factors are similar to those in area 3. An example item is “Expansion of education/development possibilities”. The response scale ranges from 1 “Only with the employer” to 5 “Only with the employee”.

### 2.2. Population, Design and Procedure

MAISE was tested in a sample of 632 employees of varying gender, age, and educational level (home care workers, industrial workers, and university employees). Data were collected from three Dutch organizations: a home care organization, an industrial organization, and a university.

In the home care organization, data were collected as part of an evaluation study of an ergonomic intervention for 500 domestic care workers. All these workers participated in the intervention. The MAISE items were included in the baseline measurement of this evaluation (January 2018). A total of 283 employees filled out the online questionnaire (response rate 57%). The MAISE items were sent to a group of 475 employees in the industrial organization by means of an online cross-sectional survey (January 2014); 205 employees responded (response rate 43%). This sample appeared to be an adequate representation of the employee population of the industrial organization (88.5% men, 38.5% older than 55 years of age, and 27.5% higher educational level). At the university, data were collected as part of an evaluation study of a lifestyle intervention. All university employees were able to participate in this intervention. The MAISE items were included in the baseline measurement of this evaluation (March 2016). Questionnaires were sent out by email to 280 respondents; 144 people filled out the questionnaire (response rate 51%). The average response rate was 50.3%. Table 1 provides an overview of the characteristics age, gender, and educational level for the total sample and the three organizations separately.

### 2.3. Ethical Issues

The following ethical measures were taken. In all three organizations, the study was approved by the HR manager or the CEO. Participants were informed by individual email and information on the organization’s intranet; employees were free to refuse to participate and welcome to ask questions and express concerns about the study. The return of a completed questionnaire was taken to imply consent. Data were stored anonymously and treated confidentially, and participant privacy was guaranteed. The study was part of a larger research project “Vitality@DSM: participation and effectiveness. A diversity-specific evaluation”, which was approved by the Medical Ethics Committee of MUMC+ (Academic Hospital Maastricht, February 2, 2012, #METC 12-4-006). 

### 2.4. Measures

In addition to the MAISE items described above, several demographics (gender, age, and educational level) and SE proxies (i.e., for testing criterion validity) were included in the online survey.

#### Proxies

Vitality was measured by means of a subscale of the Dutch version of the Utrecht Work Engagement Scale (UWES) [26,27] (5 items, Cronbach’s alpha is 0.87). The response scale ranged from 0 “never” tot 6 “always/every day”.

Health was only measured in two of the three organizations. In the industrial organization, general health was measured by means of 1 item “How would you rate your health in general?”. Response range varied from 1 “bad” to 5 “excellent”. In the university questionnaire, health was self-rated by means of 1 item (Self-Rated Health): “How would you grade your health on a scale from 1 to 10?”.

### 2.5. Data Analyses

Psychometric, correlational, and comparative (ANOVAs) analyses were performed. Construct validity of the MAISE was first examined exploratively by means of a Principal Component Analysis (PCA) with oblimin rotation. Components with an eigenvalue >1 were extracted. Items were considered indicators of the same factor if they were highly related to each other by having factor loadings higher than 0.40. These analyses were conducted by means of IBM SPSS Statistics version 25 (IBM Corp., Armonk, NY, USA). Second, a Confirmatory Factor Analysis (CFA) was performed to further validate the MAISE scales. CFA was conducted by means of JAMOVI version 0.9.5.12 [28]. JAMOVI uses the Maximum Likelihood estimation method, which is scale invariant. We constructed the models based on the PCA results. The exact fit of the model was assessed with the Chi-square index. Because of the high sensitivity of the Chi-square index to sample size [29], we used several comparative and parsimonious fit indices [30]: the RMSEA (Root Mean Square Error of Approximation, which should be lower than 0.08); the CFI (Comparative Fit Index) and the TLI (Tucker Lewis Index, also known as the Non Normed Fit Index, which should both be 0.90 or higher); and the SRMR (Standardized Root Mean Square Residual, which should be lower than 0.05). For some scales, we allowed residual errors of some items to correlate. Internal consistency was calculated by examining the Cronbach’s alpha. Criterion validity of the MAISE scales was examined by calculating the Pearson correlation coefficients among the MAISE scales and between the MAISE scales and the proxies vitality and health (the latter only in two organizations). Finally, we performed several sensitivity analyses to examine the robustness of the scales.

## 3. Results

### 3.1. Principal Component Analyses and Confirmatory Factor Analyses

Table 2 shows the results of the PCA (construct validity) and reliability analyses of the MAISE items (areas 1 and 2).

MAISE area 1 “meaning of SE” consists of two scales: (1a) fit and useful (6 items), and (1b) productive (4 items): Two factors with eigenvalue >1 were drawn, explaining 51% of total variance. Factor loadings in Table 2 are based on the pattern matrix. This factor structure was clearly confirmed in the Confirmatory Factor Analysis CFA (see Table 3). Cronbach’s alphas of both scales 1a and 1b were acceptable.

MAISE area 2 “level of SE”: initially three factors with eigenvalue >1 were drawn (2.912; 1.339; and 1.117), explaining 59.6% of total variance. The scree plot “bends” after two factors though, with the first two factors explaining 47.2% of total variance. Due to this and the theoretical notion that this area would consist of two subscales, we decided to force a 2-factor structure, which is shown in Table 2. Item 7 “I feel that I will be able to do my job until I retire” (which we initially expected to load on the factor performance), loaded higher on the factor health issues. Respondents apparently had their health in mind when answering this item and not their performance. We therefore moved this item to scale 2b. Items 3 “I enjoy my job” and 9 “It is easy for me to make money” were ambiguous and did not load high on any of the factors and were therefore deleted. Item 8 “I am rarely absent from my work due to sickness” is also not very clear, but we decided to keep this item in scale 2b for theoretical and content-related reasons.

In sum, we decided to delete items 3 and 9 and to create two scales in MAISE area 2 “my level of SE”: (2a) performance (4 items) and (2b) health issues (4 items). The Cronbach’s alpha of scale 2a was good. However, the reliability of scale 2b remained lower than we would have preferred, even after deletion of item 8. We decided to still present this scale because the two factor-structure of scale 2 was clearly confirmed in the CFA (See Table 3) and because of the low number of items in this scale. We allowed two error terms to correlate in the CFA. 

Table 4 and Table 5 show the results of the PCA (construct validity) and reliability analyses of the MAISE items of areas 3 and 5. MAISE area 3 “factors affecting my SE”: three factors with eigenvalue >1 were drawn (eigenvalues were 6.846, 4.366, and 4.780, respectively), explaining 62.5% of total variance. These scales were labelled (3a) work organization (5 items), (3b) lifestyle and balance (5 items), and (3c) adapted job (8 items) (see Table 4). Item 4 “find a better balance between my job and private life” loaded high on two factors, but we decided to keep this item in scale 3b lifestyle and balance for content-related reasons. All items in this scale refer to the behavior and situation of the individual employee.

In the CFA, this three-factor structure could not clearly be confirmed (see Table 3); most fit indices were below acceptable levels. Considering factor loadings and modification indices, we adjusted the scales by deleting several items, which resulted in more acceptable fit indices. In the adjusted version, scale 3a had 2 items, scale 3b had 5 items, and scale 3c had 6 items. The Cronbach’s alphas of all three scales (in both the original and adjusted versions) is good.

As MAISE area 4 “responsibility for SE” was measured by means of only 1 item, the factor structure was not tested. 

MAISE area 5 “responsibility for factors affecting my SE”: five factors with eigenvalues > 1 (eigenvalues are 4.277, 2.792, 1.711, 1.138, and 1.063) were drawn, explaining 61% of total variance. These scales were labelled 5a lifestyle (3 items), 5b balance (2 items), 5c adapted job (4 items), 5d work content (4 items), and 5e work context (5 items). The Cronbach’s alphas of these 5 scales range from adequate to good (see Table 5).

In the CFA, this five-factor structure was clearly confirmed (see Table 3). We allowed three error terms to correlate in the CFA. 

### 3.2. Levels of SE

Table 6 shows the means, range, standard deviations, and 25th and 75th percentiles of the scales of the MAISE in the total sample and the means, range, and standard deviations for various subgroups (age categories, gender, and educational level).

According to the employees, SE means being fit, healthy, and useful and, secondly, being productive. The latter item also refers to future productivity. According to the employees, factors affecting SE are work organization, a healthy lifestyle, a good work–life balance, and the possibility of adapting the job to their situation and wishes. Employees regard SE as a shared responsibility, with both employees and employer being equally responsible. However, employees regard employers as being more responsible for work content, work context, and adapting the job to the employee and regard themselves as more responsible for work–life balance and lifestyle.

### 3.3. Sensitivity Analyses 

The PCA and CFA were repeated for various subgroups (men vs. women, high versus lower level of education, and younger (<45 years) vs. older (≥45 years), which confirmed the robustness of the factor structures across groups (results are available on request).

We tested for differences regarding age (employees <45 years versus ≥45 years), gender and educational level (lower/middle versus higher) in the levels of the MAISE scores. With regard to age, no difference was found in the meaning of SE for employees (area 1). Regarding the description of their own level of SE (area 2), older employees scored higher on performance (F(df) = 10.88(1) *). Regarding area 3 (factors affecting SE), younger employees score higher on work organization (F(df) = 4.99(1) *) and adapted job (F(df) = 5.52(1) *). They considered these factors to be associated more with their SE than older employees did. Regarding areas 4 and 5 (responsibility for SE), in general, younger employees regarded themselves as being more responsible than their employer for their SE (F(df) = 4.30(1) #). However, looking at the specific factors, more older employees than younger employees considered themselves to be responsible for their work–life balance F(df) = 4.00 (1) #) and adapted job F(df) = 14.02 (1) #). With regard to gender, we found that men scored higher on fit and useful (F(df) = 6.54 (1) *) and productive (F(df) = 16.81 (1) *) (area 1). As regards MAISE area 2, men report slightly more health issues (F(df) = 18.90 (1) *). With regard to MAISE area 3 (factors affecting SE), women scored higher on work organization (F(df) = 20.32(1) *) and lower on lifestyle and balance (F(df) = 8.27(1) *). Regarding areas 4 and 5 (responsibility for SE), in general, men considered themselves more responsible for their SE than women (F(df) = 27.31(1) *). Looking at the specific factors, we see that men more than women considered themselves to be responsible for their lifestyle (F(df) = 6.08(1) *), while women (more than men) considered work–life balance (F(df) = 7.47(1) *) and adapted job (F(df) = 12.24(1) *) to be their responsibility. As regards educational level, we found that employees with higher educational levels scored higher on fit and useful (F(df) = 8.44(1) *) than employees with middle or lower educational levels (area 1). As regards MAISE area 2, employees with higher educational levels scored lower on performance (F(df) = 5.54(1) *) and reported more health issues (F(df) = 9.61(1) *). As regards MAISE area 3 (Factors affecting SE), employees with higher educational levels scored higher on adapted job (F(df) = 9.21(1) *). Regarding areas 4 and 5 (responsibility for SE), in general, employees with higher educational levels considered themselves as being more responsible for their SE than their employer F(df) = 18.54(1) *). Looking at specific factors, we found that employees with higher educational levels considered themselves to be responsible for their lifestyle (F(df) = 5.47(1) *) (more than employees with lower educational levels), while employees with lower educational levels considered themselves to be responsible for their work–life balance (F(df) = 9.03(1) *). Although significant, the latter differences were small. 

### 3.4. Criterion Validity 

Table 7 shows the Pearson correlation coefficients of all scales of the MAISE with the health proxies vitality and general health (for the university only) and self-rated health (for the industrial organization only) (criterion validity). As one would expect, scales 1a to 2b and particularly 2b health issues showed moderate to high and significant correlations with the health proxies vitality and general health. The scales and items of MAISE areas 4 and 5 about responsibility for SE and those factors that affect it had low correlations with the health proxies. 

## 4. Discussion

This paper describes the development and validation of the MAastricht Instrument for Sustainable Employability (MAISE). The MAISE aims to tap the employee perspective on current SE, factors affecting SE, and responsibility for SE in order to offer employers a valid needs assessment when deciding on SE interventions. The MAISE consists of 12 scales divided over five areas. We performed both exploratory and confirmatory factor analyses. Both types of analyses indicate that the MAISE is an instrument with good construct validity and adequate to good reliability in a relatively large sample including different types of employees. One scale (2b) had a relatively low reliability, but we decided to present this scale because of the clear factor-structure. Correlational analyses have shown that the criterion validity was good as well. Additionally, the factorial structure was shown to be robust across subgroups. At the same time, MAISE appears to be sensitive enough to detect mean differences between subgroups (e.g., older and younger employees).

This paper showed that employees in our research group are probably less interested in their employer offering lifestyle and work–life balance interventions as they consider themselves to be more responsible for those factors affecting SE. Rather, they would prefer their employer to improve the content and context of their work and to provide opportunities for adapting their job to their needs. This is a remarkable finding, given the situation that, in many large organizations, SE programs tend to focus on health and lifestyle and to a much lesser extent on the work situation and the job itself. This could create a “credibility gap”: interventions and programs for improving SE might not be in alignment with the perspectives and needs of the employees [14,31]. Even though, in general, employees regard SE as a shared responsibility. 

The MAISE thus focuses on measuring what the target group of employees regards as being their needs and responsibilities with regard to SE and might thus lead to interventions that would be more easily adopted by employees and more effectively implemented into the organization. Many existing SE interventions are ineffective because of implementation failure [32]. The MAISE approach is different, in the sense that it does not merely target determinants of SE, as is promoted in the Intervention Mapping approach, for example [33]. This explicit link to implementation is certainly the added value of the current instrument when compared with existing SE instruments [1,2,8,20]. However, the employee opinion on factors affecting their SE and the determinants that have been statistically proven might not be completely parallel.

The MAISE appears to be able to detect age-related differences regarding SE, associated factors, and responsibilities. The scores regarding the meaning attached to SE by employees are rather stable across the two age groups (younger and older employees). Older employees scored somewhat higher on performance, and work organization and job adaptations were less important to them as factors affecting their SE compared with younger employees. Older employees might have more working experience and thus more autonomy to navigate in the organization themselves [34]. In general, older employees hold themselves less responsible for their SE than younger employees. Young people entering the labor market might be more aware of their own responsibility for employability due to governmental and organizational attention for this issue and might have less job security. However, considering the responsibility for specific factors, older employees regard themselves as being responsible for their work–life balance and job adaptations.

### 4.1. Recommendations for Future Use of the MAISE

In general, the MAISE can be used for a needs assessment at group level (and the scores found in this study can be used as benchmarks or norm scores). When SE interventions become better aligned with the employee perspective, they will probably be more effective. Organizations interested in maintaining and improving employee SE could first use the scales from MAISE areas 1, 3, 4, and 5 in order to give “voice” to employees, and choose/develop interventions alike to improve SE. The effects of the interventions could be evaluated using the scales from areas 2 (level of SE) and 3 (factors affecting SE). The MAISE scales from area 2 could also be used in long-term research regarding employees’ job success and ability to maintain employed until retirement within the same or in another organization.

### 4.2. Study Limitations and Future Research

This study builds on a representative sample from a varied population of employees working in the industrial sector, health care, and a university setting. The sample included employees of varying ages, gender, and educational levels. The average response rate was 50.3%, which is relatively high given the setting [20]. We will certainly have missed employees who are illiterate or unable to read Dutch sufficiently well. Finally, the results may be influenced by some forms of common method variance or artificial inflation of synchrony in answers, which is inherent to all self-report and cross-sectional data [35]. Further validation of the MAISE in larger samples from various sectors of work and in vulnerable groups of workers is recommended. To reach all employees, versions translated to an immigrant employee’s native language and a simplified version should also be developed and validated. In conclusion, research on the relationship between SE determinants, the employee opinion on factors affecting their SE, and the effectiveness of SE interventions informed by MAISE outcomes is needed.

## 5. Conclusions

The MAastricht Instrument for Sustainable Employability (MAISE) appears to be a valid instrument for tapping the employee perspective on current SE, factors affecting SE, and responsibility for SE. The MAISE consists of 12 scales divided over five areas and has a good construct and criterion validity and an adequate to good reliability. The MAISE appears to be able to detect age-related differences regarding SE, associated factors, and responsibilities. In general, the MAISE can be used by employers to better align SE interventions with the employee perspective.

## Figures and Tables

**Table 1 ijerph-17-02211-t001:** Sample characteristics: mean age, gender (%), and educational level (%).

Variable	Total Sample	Industry	University	Homecare
Age (mean)	48.1	49.2	40.1	51.2
Gender (%)				
- men	38.7	87.3	20.8	4.4
- women	61.3	12.7	79.2	95.6
Educational level (%)				
- primary education	3.6	0.5	0	8.3
- secondary education	31.5	17.1	10.4	55.9
- medium professional education, lower levels	9.3	12.2	0.8	11.4
- medium professional education, higher levels	23.3	42.9	5.6	15.3
- higher professional education and university	32.4	27.3	83.2	9.2

**Table 2 ijerph-17-02211-t002:** Principal Component Analysis (PCA) MAastricht Instrument for Sustainable Employability (MAISE) areas: (1) meaning of (sustainable employability) SE and (2) my level of SE, oblimin rotation.

	Sustainable employability has the following meaning to me:		
#	Item	Fit and Useful	Productive
1	I can do my job without too much stress	**0.721**	−0.107
2	I have the right knowledge to perform my job well	**0.657**	0.091
3	I enjoy my job	**0.827**	−0.085
4	I do not develop physical health issues as a result of my job	**0.593**	0.175
5	The capacity to do my job efficiently	**0.537**	0.314
6	The feeling of performing useful activities	**0.727**	0.009
	Cronbach’s alpha scale 1a fit and useful	0.80	
7	Being able to do my work until I retire	0.134	**0.599**
8	Try to keep my absenteeism limited	−0.194	**0.898**
9	I can make money	0.043	**0.534**
10	I am productive while working	0.204	**0.607**
	Cronbach’s alpha scale 1b productive		0.65
**2**	To what extent do the following statements apply to you?		
#	Item	Performance	Health issues
2	I have the required knowledge to perform my job well	**0.783**	−0.153
3	I enjoy my job		
5	I am efficient at my job	**0.851**	−0.061
6	I feel that my job activities are useful	**0.772**	0.059
10	I am productive while working	**0.785**	0.077
	Cronbach’s alpha scale 2a performance (2–5–6–10)	0.81	
1	My job is stressful	−0.106	**0.638**
4	I have work-related physical health issues	−0.064	**0.829**
7	I feel that I will be able to do my job until I retire	0.283	**0.524**
8	I am rarely absent from work due to sickness	0.281	**0.244**
9	It is easy for me to make money		
	Cronbach’s alpha scale 2b health issues (1–4–7)		0.46
	Cronbach’s alpha scale 2b health issues (1–4–7–8)		0.45

**Table 3 ijerph-17-02211-t003:** Fit indices of the MAISE areas.

		*Chi-2 (df)*	*CFI*	*TLI*	*SRMR*	*RMSEA*
1	Meaning of SE (2 factors)	91.8 (33)	0.966	0.954	0.034	0.054
2	Level of SE (2 factors)	70.87 (17)	0.949	0.916	0.046	0.073
3	Factors affecting SE (3 factors)	886 (128)	0.873	0.848	0.118	0.102
3	Factors affecting SE—adjusted (3 factors)	300 (61)	0.939	0.922	0.048	0.083
4	Responsibility for factors affecting SE (5 factors)	364 (122)	0.927	0.909	0.050	0.056

*Note.* CFI = Comparative Fit Index; TLI = Tucker Lewis Index; SRMR = Standardized Root Mean Square Residual; RMSEA = Root Mean Square Error of Approximation.

**Table 4 ijerph-17-02211-t004:** PCA and reliability analysis MAISE area (3) factors affecting my SE, oblimin rotation.

3 Indicate to what extent you believe the following changes could contribute to YOUR sustainable employability:				
#	Item	Work Organization	Lifestyle and Balance	Adapted Job
1	Move more	0.095	**0.727**	0.073
2	Reach a healthier body weight	0.014	**0.920**	−0.117
3	Start eating more healthy	−0.020	**0.914**	−0.024
4	Find a better balance between my job and private life	−0.001	**0.469**	**0.488**
5	Learn to deal with stress better	0.220	**0.425**	0.317
	Cronbach’s alpha scale lifestyle and balance		0.85	
	Cronbach’s alpha scale lifestyle and balance—adjusted		0.90	
6	Decrease in job pressure	0.189	0.331	**0.398**
7	Introduce more flexible working hours	0.131	0.081	**0.695**
8	More attention paid to career paths	0.394	−0.056	**0.538**
9	Reducing weekly working hours	−0.175	0.044	**0.862**
10	Change of job tasks/function/activities	−401	−0.177	**0.573**
	Cronbach’s alpha scale adapted job			0.81
11	Atmosphere improvement within my department/team	**0.630**	0.031	0.047
12	Improvement of working conditions	**0.612**	0.053	0.147
13	Expansion of education/development possibilities	**0.665**	0.025	0.091
14	More variation in job activities	**0.849**	0.003	−0.037
15	More challenging job activities	**0.884**	−0.046	−0.057
16	To receive more appreciation for the job that I do	**0.756**	0.081	−0.131
17	The chance to apply my knowledge/skillset to my job better	**0.802**	0.044	0.044
18	Obtain more responsibility within my job	**0.833**	0.004	−0.036
	Cronbach’s alpha scale work organization	0.90		
	Cronbach’s alpha scale work organization—adjusted	0.87		

**Table 5 ijerph-17-02211-t005:** PCA and reliability analysis MAISE area (5) responsibility for factors affecting my SE, oblimin rotation.

5	Indicate where you feel the responsibility lies for implementing the changes below that would improve YOUR sustainable employability:					
#	Item	Work Content	Lifestyle	Adapted Job	Work Context	Balance
1	Move more	−0.071	**0.812**	0.016	0.040	0.089
2	Reach a healthier body weight	0.011	**0.931**	−0.011	−0.057	0.033
3	Start eating more healthy	−0.029	**0.931**	−0.011	−0.057	−0.033
	Cronbach’s alpha scale responsibility for lifestyle		0.88			
4	Find a better balance between my job and private life	0.037	0.150	0.149	0.045	**0.686**
5	Learn to deal with stress better	0.121	0.171	0.047	0.036	**0.703**
	Cronbach’s alpha scale responsibility for balance					0.60
7	Introduce more flexible working hours	−0.160	−0.057	**0.634**	0.293	0.119
8	More attention paid to career paths	0.066	0.010	**0.659**	0.012	0.055
9	Reducing weekly working hours	−0.133	0.061	**0.817**	−0.0090	0.041
10	Change of job tasks/function/activities	0.352	-0.031	**0.692**	−0.137	−0.036
	Cronbach’s alpha scale responsibility for adapted job			0.71		
6	Decrease in job pressure	−0.022	−0.152	0.201	**0.594**	0.362
11	Atmosphere improvement within my department/team	0.167	0.131	−0.147	**0.433**	0.089
12	Improvement of working conditions	−0.080	−0.041	0.034	**0.814**	0.043
13	Expansion of education/development possibilities	0.048	0.117	0.219	**0.548**	−0.409
14	More variation in job activities	**0.431**	0.048	0.280	0.314	−0.164
15	More challenging job activities	**0.494**	0.114	0.216	0.290	−0.236
16	To receive more appreciation for the job that I do	0.120	−0.067	−0.075	**0.613**	−0.090
17	The chance to apply my knowledge/skillset to my job better	**0.798**	−0.026	−0.061	0.030	0.054
18	Obtain more responsibility within my job	**0.850**	−0.047	−0.004	−0.033	0.131
	Cronbach’s alpha scale responsibility for work content (items 14–15–17–18)	0.76				
	Cronbach’s alpha scale responsibility for work context (items 6–11–12–13–16)				0.66	

**Table 6 ijerph-17-02211-t006:** Means (M), standard deviations (SD), and percentiles of the MAISE scales for the total sample and subgroups.

Scale (range 1–5)	#	M (range)	SD	25th perc.	75th perc.	M (range)	SD	M (range)	SD	M (range)	SD	M (range)	SD	M (range)	SD	M (range)	SD
		Total sample (*n* = 601)				<45 (*n* = 359)		≥45 (*n* = 177)		Men (*n* = 202)		Women (*n* = 339)		Lower/middle ed. (*n* = 365)		Higher ed. (*n* = 179)	
**1. Meaning of SE**																	
1a. Fit and useful	6	4.16 (1–5)	0.48	4.00	4.40	4.15 (1–5)	0.49	4.19 (2.83–5)	0.44	4.23 (2.67–5)	0.43	4.12 (1–5)	0.49	4.12 (1–5)	0.48	4.25 (3–5)	0.42
1b. Productive	4	4.01 (1.5–5)	0.53	3.75	4.50	4.08 (1.5-5)	0.52	4.13 (2.75–5)	0.50	4.21 (2.5–5)	0.52	4.02 (1.5–5)	0.51	4.07 (1.5–5)	0.54	4.16 (3–5)	0.47
**2. Level of SE**																	
2a. Performance	4	4.11 (1–5)	0.49	4.00	4.25	4.07 (1–5)	0.49	4.21 (2.5–5)	0.49	4.06 (2.5–5)	0.41	4.14 (1–5)	0.54	4.15 (1–5)	0.48	4.04 (2.5–5)	0.52
2b. Health issues	4	3.69 (1.75–5)	0.57	3.25	4.00	3.71 (1.75–5)	0.57	3.71 (2.25–5)	0.57	3.84 (2.25–5)	0.53	3.62 (1.75–5)	0.58	3.65 (1.75–5)	0.56	3.81 (1.75–5)	0.58
**3. Factors affecting SE**																	
3a. Work organization	2	3.06 (1–5)	0.85	2.50	3.67	3.11 (1–5)	0.82	2.94 (1–4.67)	0.85	2.86 (1–5)	0.87	3.18 (1–5)	0.79	3.02 (1–5)	0.81	3.15 (1–5)	0.89
3b. Lifestyle and balance	5	2.91 (1–5)	1.09	2.00	4.00	2.95 (1–5)	1.08	2.86 (1–5)	1.08	3.09 (1–5)	1.01	2.82 (1–5)	1.11	2.94 (1–5)	1.09	2.88 (1–5)	1.07
3c. Adapted job	6	2.85 (1–5)	0.84	2.20	3.40	2.91 (1–5)	0.79	2.74 (1–5)	0.88	2.88 (1–5)	0.82	2.86 (1–5)	0.84	2.79 (1–5)	0.84	3.02 (1–5)	0.80
**4. Responsibility for employee SE**																	
Who is responsible for employee SE	1	2.86 (1–4)	0.47	3.00	3.00	2.90 (1–4)	0.42	2.81 (1–4)	0.55	3.00 (2–4)	0.30	2.78 (1–4)	0.53	2.81 (1–4)	0.52	2.99 (2–4)	0.32
**5. Responsibility for factors affecting SE**																	
5a. Life style	3	4.13 (1–5)	0.68	3.67	4.67	4.16 (2–5)	0.66	4.10 (1–5)	0.70	4.23 (1–5)	0.64	4.08 (2–5)	0.69	4.09 (2–5)	0.96	4.23 (1–5)	0.63
5b. Balance	2	3.50 (1–5)	0.63	3.00	4.00	3.47 (1–5)	0.63	3.51 (2–5)	0.64	3.41 (2–5)	0.60	3.56 (1–5)	0.65	3.56 (1–5)	0.68	3.39 (2–5)	0.53
5c. Adapted job	4	2.79 (1–5)	0.56	2.50	3.00	2.72 (1–4.5)	0.51	2.91 (1–5)	0.64	2.68 (1–4.5)	0.52	2.86 (1–5)	0.57	2.81 (1–5)	0.60	2.75 (1–4.5)	0.45
5d. Work content	4	2.81 (1–5)	0.55	2.50	3.00	2.82 (1–5)	0.52	2.81 (1–5)	0.60	2.76 (1–5)	0.53	2.84 (1–5)	0.55	2.79 (1–5)	0.59	2.85 (1.5–4.5)	0.43
5e. Work context	5	2.55 (1–4.6)	0.46	2.20	2.80	2.52 (1–4.6)	0.46	2.60 (1.2–4)	0.45	2.54 (1–3.8)	0.47	2.56 (1–4.6)	0.46	2.55 (1–4.6)	0.48	2.54 (1.2–3.6)	0.41

*Note.* A higher score reflects a more positive score on the particular variable, except for the “health issues” subscale: here, a higher score reflects more health problems. A higher score on scale 3 means that this particular factor contributes a lot to SE. A higher score on scales 4 and 5 means that responsibility lies mainly with the employee. The bold mean scores indicate that the subgroups differ significantly on this scale (i.e., young vs. old, men vs. women, and lower/middle vs. higher educated).

**Table 7 ijerph-17-02211-t007:** Pearson correlations MAISE scales and items and proxies (N ranges from 128 to 601).

#	Variable ^a^	1a	1b	2a	2b	3a	3b	3c	4	5a	5b	5c	5d	5e	6	7	8
	***MAISE scales***																
1a	Useful	--															
1b	Prod.	0.62 **	--														
2a	Perf.	0.43 **	0.34 **	--													
2b	Health	0.33 **	0.38 **	0.32 **	--												
3a	Work org.	0.04	0.06	−0.02	−0.17 **	--											
3b	Lifestyle	−0.00	0.12 **	−0.02	−0.02	0.34 **	--										
3c	Adapted	0.05	0.04	−0.10 *	−0.22 **	0.64 **	0.38 **	--									
4	Resp. SE	0.05	0.00	−0.13 **	0.06	−0.02	0.12 **	0.01	--								
5a	Life.-res.	0.10 *	0.10 *	−0.04	0.06	−0.05	−0.09 *	−0.08	0.16 **	--							
5b	Bal.-res.	−0.00	0.02	0.10*	0.03	−0.06	−0.06	−0.16 **	−0.02	0.38 **	--						
5c	Adap.-res.	−0.05	−0.06	0.06	0.03	−0.06	−0.06	−0.24 **	0.05	0.06	0.24 **	--					
5d	Content-res.	−0.11 **	−0.13 **	−0.05	−0.02	−0.10 *	−0.01	−0.13 **	0.04	0.08	0.12 **	0.32 **	--				
5e	Context-res.	−0.12 **	−0.08	−0.06	0.07	−0.11 *	−0.03	−0.11 *	0.09*	−0.00	0.12 **	0.39 **	0.53 **	--			
	***Health Proxies***																
6	Vit.	0.19 **	0.13 **	0.41 **	0.33 **	−0.02	−0.02	−0.18 **	−0.08 *	−0.01	0.08	0.14 **	0.05	0.01	--		
7	Health	0.25 **	0.19*	0.20 *	0.52 **	−0.11	−0.10	−0.14	−0.01	−0.08	−0.11	−0.05	0.14	−0.03	0.36 **	--	
8	SRH	0.24 **	0.11 *	0.18 **	0.41 **	−0.05	−0.21 **	−0.05	−0.06	0.00	−0.02	0.00	0.02	0.05	0.34 **	0.95 **	--

* *p* < 0.05; ** *p* < 0.01. ^a^ Explanation of variable names: Useful = Fit and useful; Prod. = Productive; Perf. = Performance; Health = Health issues; Work org. = Work organization; Lifestyle = Lifestyle and balance; Adapted = Adapted job; Resp. SE = Overall responsibility for SE; Life.-res = Responsibility for lifestyle; Bal.-res = Responsibility for balance; Adap.-res. = Responsibility for adapted job; Content-res. = Responsibility for work content; Context-res. = Responsibility for work context; Vit. = Vitality (engagement); Health = General health (only for university); SRH = Self Rated Health (only for industrial organization).

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
