# Peer review of "Tapping the Employee Perspective on the Improvement of Sustainable Employability (SE): Validation of the MAastricht Instrument for SE (MAISE-NL)"

_ijerph, 2020, doi:10.3390/ijerph17072211_

Round 1
Reviewer 1 Report
Table 1 is confused. Information about the total sample, ind, univ and homecare are not clear. According to the text above the table, numbers meaning % of response. In this case, add "% of response" above the "Ind. Univ. Homecare". In additon, at the end of the table, add the description of ind and univ.
Consider the possibility of add a short table of descriptive data with the mean, median, min, max and range for the results of the difference scales and sub scales. Because the classification of gender and education are shown, descriptive results could improve the article. I can see information in table 6, but is for the total sample. If the information is divided, you can show more details about the sample.
Also, information about the Cronbach alpha is confused. For example, in Table 2 Cronbach alpha for fit & useful and productive sub scales are in the same column. Just move the value to the best column. The same issue for table 4 and 5.
The study is complete and show the validity of a are instrument. However, I would like to see a more extensive classification of the information to ensure the validation is enough.
Probably authors are considering to develop a structural equation model, not in this article, but in a near future.
Author Response
Dear Jovana Kutorov and dear reviewers,
Thank you very much for giving us the opportunity to resubmit the manuscript entitled “Tapping the employee perspective on the improvement of Sustainable Employability (SE): Validation of the MAastricht Instrument for SE (MAISE-NL)” (ID ijerph-739343). We would like to thank the reviewers for their comments to our manuscript. We have revised the manuscript in accordance with these comments, and we have marked the revisions in the manuscript by means of the "Track Changes" function in Microsoft Word. We have also performed a final spell check. Below we have indicated (in Italics) how we have responded to the reviewers’ comments.
We look forward to your decisions regarding the revised manuscript. We hope you find the revisions appropriate, the article improved and worthy to be published in IJERPH. Should you have any questions or comments about the revisions, please do not hesitate to contact us.
Kind regards, also on behalf of the co-authors,
Inge Houkes
Reviewer 1
Table 1 is confused. Information about the total sample, ind, univ and homecare are not clear. According to the text above the table, numbers meaning % of response. In this case, add "% of response" above the "Ind. Univ. Homecare". In additon, at the end of the table, add the description of ind and univ.
Dear reviewer, thank you for this comment. We apologize for the unclarities in Table 1. We also noticed that the table title was not clear. Please note that the table does not depict response rates, but shows the sample characteristics mean age, gender division (in %), and level of education (in %). We have made the following adjustments: we have clarified the text above the table (lines 195-196), we have adjusted the table title (line 196), and we have made several adjustments in the table itself (we have now written the abbreviations ind. and univ. in full and we have deleted the Dutch abbreviations MBO1-2 and MBO3-4). We hope that it is now clear that age is depicted in mean score of years, and gender and educational level in percentages.
Consider the possibility of add a short table of descriptive data with the mean, median, min, max and range for the results of the difference scales and sub scales. Because the classification of gender and education are shown, descriptive results could improve the article. I can see information in table 6, but is for the total sample. If the information is divided, you can show more details about the sample.
Dear reviewer, based on your suggestion we have performed several additional sensitivity analyses and we have computed several descriptives for gender and educational level as well (in addition to age categories). We have added this information to table 6 and in the section about sensitivity analyses (lines 323-342).
Also, information about the Cronbach alpha is confused. For example, in Table 2 Cronbach alpha for fit & useful and productive sub scales are in the same column. Just move the value to the best column. The same issue for table 4 and 5.
Dear reviewer, thank you for this comment. We have moved the Cronbach’s alpha for all subscales to the best fitting column in Tables 2, 4 and 5. We have also restored all table titles (we assume that in the editing of the manuscript the first words of all table titles disappeared).
The study is complete and show the validity of a are instrument. However, I would like to see a more extensive classification of the information to ensure the validation is enough.
Dear reviewer, we have inserted a more extensive classification of the study findings in the first paragraph of the Discussion section (lines 365-372).
Probably authors are considering to develop a structural equation model, not in this article, but in a near future.
Dear reviewer, thank you for this suggestion. We are indeed considering to develop a structural equation model for the MAISE and determinants thereof in future studies.
Reviewer 2 Report
SE should be referred to as a "construct", not "concept"
In Section 1.1 is merely definitional. I would suggest giving 1 or 2 practical examples of this construct.
In Section 1.2, I would like to have seen a better fleshing out of the construct relationship with employee engagement. Is SE similar to employee engagement, only potentially across jobs and organizations? Is SE more of an individual-level (personality) variable as opposed to a contextual or situational one? Of so, this should be better explained. Also, how does SE relate to work ethic?
Is there a relation between SE and (predicted) tenure with an organization?
Could the MAISE be adapted and used in assessing job success and tenure with an organization?
Author Response
Dear Jovana Kutorov and dear reviewers,
Thank you very much for giving us the opportunity to resubmit the manuscript entitled “Tapping the employee perspective on the improvement of Sustainable Employability (SE): Validation of the MAastricht Instrument for SE (MAISE-NL)” (ID ijerph-739343). We would like to thank the reviewers for their comments to our manuscript. We have revised the manuscript in accordance with these comments, and we have marked the revisions in the manuscript by means of the "Track Changes" function in Microsoft Word. We have also performed a final spell check. Below we have indicated (in Italics) how we have responded to the reviewers’ comments.
We look forward to your decisions regarding the revised manuscript. We hope you find the revisions appropriate, the article improved and worthy to be published in IJERPH. Should you have any questions or comments about the revisions, please do not hesitate to contact us.
Kind regards, also on behalf of the co-authors,
Inge Houkes
Reviewer 2
SE should be referred to as a "construct", not "concept"
In Section 1.1 is merely definitional. I would suggest giving 1 or 2 practical examples of this construct.
Dear reviewer, thank you for this comment. We have replaced the term ‘concept’ by ‘construct’ throughout the manuscript (in reference to SE). We have also added an example of SE in section 1.1 (lines 62-65).
In Section 1.2, I would like to have seen a better fleshing out of the construct relationship with employee engagement. Is SE similar to employee engagement, only potentially across jobs and organizations? Is SE more of an individual-level (personality) variable as opposed to a contextual or situational one? Of so, this should be better explained. Also, how does SE relate to work ethic?
Dear reviewer, you ask for a better explanation of the distinction between the SE construct and related concepts such as employee engagement and work ethic. We decided to provide more information about this in section 1.1, lines 53-57 (as section 1.2 primarily focuses on SE interventions). We feel that journal space does not permit to elaborate more extensively about the SE construct in the manuscript, but for a more detailed description of the SE construct and its relation with other concepts we refer to Fleuren et al. (2016), Van der Klink et al. (2016), and Hazelzet et al. (2019).
Is there a relation between SE and (predicted) tenure with an organization?
Dear reviewer, thank you for this interesting question. The construct SE in itself entails a longterm perspective, and implies that – strictly speaking - SE should not be measured at one moment in time. We did not validate the MAISE in relation to (predicted) tenure, but in future studies this should certainly be addressed. We have described this option for future research in the Discussion section (lines 409-411).
Could the MAISE be adapted and used in assessing job success and tenure with an organization?
Dear reviewer, this could be an interesting option for future (long term) research. Please also see our response to your previous comment. Particular scales of the MAISE could be used for this purpose: are employees who score high on SE more successful in their jobs and are they staying with an organization until retirement? See lines 409-411 of the manuscript.
Reviewer 3 Report
The health of the employees is considering as one of the most important factors, which influence corporate effectiveness. The authors want to define complex factors as a potential tool for the evaluation of physical and psychometric qualities. To reach defined aims (L 122-124) they used Factor analysis with Principal Component analysis. For the verification of defined factors, they applied Cronbach's alpha ratio. This ratio must be over 0,5 if the factors could be accepted. In table 2 (L235) is the value of Cronbach's alpha under 0,5 and there is also no description, why they present it.
From the global point of view, the paper could be considered as a good average. After the correction, it could be accepted for publishing.
Author Response
Dear Jovana Kutorov and dear reviewers,
Thank you very much for giving us the opportunity to resubmit the manuscript entitled “Tapping the employee perspective on the improvement of Sustainable Employability (SE): Validation of the MAastricht Instrument for SE (MAISE-NL)” (ID ijerph-739343). We would like to thank the reviewers for their comments to our manuscript. We have revised the manuscript in accordance with these comments, and we have marked the revisions in the manuscript by means of the "Track Changes" function in Microsoft Word. We have also performed a final spell check. Below we have indicated (in Italics) how we have responded to the reviewers’ comments.
We look forward to your decisions regarding the revised manuscript. We hope you find the revisions appropriate, the article improved and worthy to be published in IJERPH. Should you have any questions or comments about the revisions, please do not hesitate to contact us.
Kind regards, also on behalf of the co-authors,
Inge Houkes
Reviewer 3
The health of the employees is considering as one of the most important factors, which influence corporate effectiveness. The authors want to define complex factors as a potential tool for the evaluation of physical and psychometric qualities. To reach defined aims (L 122-124) they used Factor analysis with Principal Component analysis. For the verification of defined factors, they applied Cronbach's alpha ratio. This ratio must be over 0,5 if the factors could be accepted. In table 2 (L235) is the value of Cronbach's alpha under 0,5 and there is also no description, why they present it.
From the global point of view, the paper could be considered as a good average. After the correction, it could be accepted for publishing.
Dear reviewer, thank you for this comment. Cronbach’s alpha of scale 2b is indeed lower than 0.5. In lines 262-267 of the manuscript we describe why we presented this scale anyway: “The reliability of scale 2b remained lower than we would have preferred, even after deletion of item 8. We decided to maintain this scale because the two factor-structure of scale 2 was clearly confirmed in the CFA (See Table 3), and because of the low number of items in this scale.” We also added a remark about this in the Discussion section (lines 367-369).
Round 2
Reviewer 1 Report
Dear authors
Thank you very much for attending and considering the suggestions I made. Please clean the final manuscript before submitting the final version.